# Grey Wolf Optimizer for Variable Selection in Quantification of Quaternary Edible Blend Oil by Ultraviolet-Visible Spectroscopy

**DOI:** 10.3390/molecules27165141

**Published:** 2022-08-12

**Authors:** Rongling Zhang, Xinyan Wu, Yujie Chen, Yang Xiang, Dan Liu, Xihui Bian

**Affiliations:** 1State Key Laboratory of Separation Membranes and Membrane Processes, School of Chemical Engineering and Technology, Tiangong University, Tianjin 300387, China; 2Key Lab of Process Analysis and Control of Sichuan Universities, Yibin University, Yibin 644000, China; 3State Key Laboratory of Plateau Ecology and Agriculture, Qinghai University, Xining 810016, China

**Keywords:** edible blend oil, spectral analysis, variable selection, multivariate calibration, grey wolf optimizer

## Abstract

A novel swarm intelligence algorithm, discretized grey wolf optimizer (GWO), was introduced as a variable selection tool in edible blend oil analysis for the first time. In the approach, positions of wolves were updated and then discretized by logical function. The performance of a wolf pack, the iteration number and the number of wolves were investigated. The partial least squares (PLS) method was used to establish and predict single oil contents in samples. To validate the method, 102 edible blend oil samples containing soybean oil, sunflower oil, peanut oil and sesame oil were measured by an ultraviolet-visible (UV-Vis) spectrophotometer. The results demonstrated that GWO-PLS models can provide best prediction accuracy with least variables compared with full-spectrum PLS, Monte Carlo uninformative variable elimination-PLS (MCUVE-PLS) and randomization test-PLS (RT-PLS). The determination coefficients (R^2^) of GWO-PLS were all above 0.95. Therefore, the research indicates the feasibility of using discretized GWO for variable selection in rapid determination of quaternary edible blend oil.

## 1. Introduction

Edible oils have been an indispensable component of our diet in everyday life, since they contain various nutritional components that are necessary for the body [1]. A variety of edible oils exist in the commercial oil market, such as soybean oil, sunflower oil, peanut oil, sesame oil, olive oil, etc. However, most single oils are unable to meet all nutritional requirements. Therefore, edible blend oil, composed of two or more single oils, may be favored by humans because of its more balanced nutritional quality, delicious taste and delightful odor [2,3]. Due to the greatly varied prices and nutritional values of different single oils, mislabeling of the oil contents for economic profit, particularly overstated labeling of those that are high-priced is a serious problem [4]. This phenomenon brings great harm not only to the health and security of consumers but also to the economic interests of manufacturers [5]. However, it is very difficult to determine the single oil contents owing to similar physical and chemical properties. Therefore, there is an urgent need to develop quantitative analysis methods for the contents of single oils in edible blend oil.

Many efforts have been devoted to the exploitation and application of analytical techniques for edible blend oil [6,7,8]. Among these techniques, ultraviolet-visible (UV-Vis) spectroscopy has proved to be a powerful tool, because it is stable, inexpensive, non-destructive and relatively fast [9,10]. An UV-Vis spectrophotometer is extraordinarily common in laboratories and widely used. However, it is difficult to direct quantitative analysis of samples via UV-Vis spectroscopy, owing to strong spectral overlapping bands. Multivariate calibration is a well-established and extensively used method in quantitation of samples, including artificial neural network (ANN) [11], support vector regression (SVR) [12], extreme learning machine (ELM) [13], principal component regression (PCR) [14] and partial least squares (PLS) [15], etc. Among these methods, PLS has drawn more attention for its practicability and versatility. However, the original UV-Vis spectra contain some irrelevant variables, which may influence the model performance, in terms of accuracy, during the modeling process.

Variable selection is quite a significant step in multivariate analysis, which can obtain good prediction results by eliminating unnecessary variables and reducing the model complexity [16,17,18]. Many variable selection methods are constantly emerging. Successive projections algorithm (SPA) has great superiority in solving the high collinearity problem. However, the number of selected variables in SPA should be smaller than the number of calibration samples [19]. The competitive adaptive reweighted sampling (CARS) method adopts an exponentially decreasing function (EDF) to incessantly shrink the variable space. Though CARS is fast, the stability of the model is low [20]. Uninformative variable elimination (UVE) eliminates variables according to the stability of each variable calculated by leave-one-out cross validation. The procedure is time-consuming for a large data set [21]. Monte Carlo uninformative variable elimination (MCUVE) overcomes the shortcoming of UVE, which has high stability and fast computation, since extra random variables are not employed. However, MCUVE tends to select more variables [22,23]. Randomization test (RT) is simple and effective for spectral variable selection, but the information of response, rather than the combination effect of variables is considered [24]. Swarm intelligence optimization algorithm (SIOA), based on the wisdom of crowd effect shows obvious merits of intelligibility, adaptivity and self-organization [25].

In recent years, grey wolf optimizer (GWO), which was initially proposed by Mirjalili et al., has become the focus of attention, due to its strong global optimization capabilities [26]. It simulates the leadership hierarchy and group hunting mechanism of wolves, making the population gradually move towards global optimization. Due to the advantages of its simple structure, fast convergence and less control parameters, GWO has been widely applied in the fields of food [27], medicine [28], industry [29,30] and agriculture [31,32,33]. Unfortunately, to date, GWO coupled with UV-Vis spectroscopy for determination of single oils contents in edible blend oil, has not been reported.

A new meta-heuristic algorithm GWO was introduced and discretized as a variable selection in spectroscopy for quaternary edible blend oil samples. The performance of wolf pack, the number of iterations and wolves, and the efficiency of GWO were investigated. The PLS model was employed to determine single oil contents. The root mean square error of cross validation (RMSECV) was used to evaluate the proposed method. The results showed that GWO-PLS can obtain the highest efficiency in elimination of uninformative variables, as well as improvement of the predictive accuracy of models, compared with other variable selection methods. Consequently, it is a promising and effective tool for quantitative analysis of quaternary edible blend oil.

## 2. Materials and Methods

### 2.1. Materials and Apparatus

Four kinds of edible oils within their expiration date, including soybean oil, sunflower oil, peanut oil and sesame oil, were purchased from supermarkets in Tianjin and utilized to prepare the quaternary edible blend oil. Soybean oil, sunflower oil and peanut oil are generally employed for cooking and frying, and have relatively low prices. Sesame oil, with a high price, is commonly used for cold dishes in China because of its strong nutty aroma and taste. To prevent changes in the chemical composition, all samples were stored in a refrigerator at 4 °C prior to analysis. An UV-Vis spectrophotometer (Evolution300, Thermo Fisher Company, Waltham, MA, USA) was employed for spectral measurement. Analytical balance (FA2004, Shanghai Sunning Hengping Scientific Instrument Co. Ltd., Shanghai, China) was used to weigh the samples. An ultrasonic instrument (SK6200HP, Kudos Ultrasonic Instrument Company, Shanghai, China) was applied to mix samples equality.

### 2.2. Sample Preparation

To evaluate the feasibility of GWO-PLS, 102 samples were prepared by mixing soybean oil with sunflower oil, peanut oil and sesame oil to form quaternary edible blend oils. The content of soybean oil was in the range of 0–100% (g/g) with an interval of ca. 2%. The contents of sunflower oil and peanut oil were in the range of 0–98% (g/g) and 0–96% (g/g), respectively. The content of sesame oil was in the range of 0–10% (g/g) with an interval of ca. 0.2%. The concentration statistics of training and prediction sets are listed in Table 1. All samples were shaken gently and then put in the ultrasonic instrument for 30 min to ensure the homogeneity of oils and to remove bubbles in samples.

### 2.3. Spectral Measurements

An UV-Vis spectrophotometer (Evolution300, Thermo Fisher Company, Waltham, MA, USA) was used to measure the spectra of edible blend oil samples. Each spectrum was composed of 601 variables recorded in the wavelength range 200–800 nm with an interval of 1 nm. A quartz cuvette with optical path of 1 cm was selected during spectral acquisition. Figure 1 shows the UV-Vis spectra of 102 quaternary edible blend oil samples. As can be seen from Figure 1, obvious noise could be found in 200–350 nm. The absorption peaks were mainly located at 427 nm, 452 nm and 478 nm, which were mainly due to the triglycerides, fatty acids and fat-soluble vitamins in edible blend oil. The peak in the region of 420–450 nm was attributed to the -CH=CH- functional groups of unsaturated fatty acids. However, it was impossible to establish a one-to-one correspondence between observed feature peaks and oil components because it was difficult to distinguish the fatty acids in all oils. Thus, chemometric methods were needed to quantitatively analyze the quaternary edible blend oil samples.

### 2.4. Chemometric Tools and Evaluation

#### 2.4.1. Grey Wolf Optimizer

GWO is inspired by observation of grey wolves living in nature. What is particularly interesting is that they have a very strict social hierarchy. The leader is the alpha (α) wolf, responsible for decision-making in the group, followed by beta (β), delta (δ) and omega (ω) wolves. Moreover, pack hunting is another indispensable group behavior of grey wolves. The hunting behavior mainly includes three stages, i.e., tracking and approaching the prey, encircling and harassing the prey until it stops moving and attacking the prey.

In the whole process, α, β and δ are denoted as the best solution, the second-best solution and the third best solution, respectively, and ω wolves are considered to be the rest solution in a pack. The mathematical model of encircling prey is formulated as follows:(1)$$\vec{D}=\left|\vec{C}\cdot \vec{Xp}(t)-\vec{X}(t)\right|$$
(2)$$\vec{X}(t+1)=\vec{Xp}(t)-\vec{A}\cdot \vec{D}$$
where $\vec{D}$ represents the estimated distance vectors between grey wolf and the prey, *t* is the current iteration. $\vec{Xp}$ indicates the position vector of the prey and $\vec{X}$ is the position vector of a grey wolf. $\vec{A}$ and $\vec{C}$ are the coefficient vectors. $\vec{A}$ is a convergence coefficient that can keep balance between exploration and exploitation.

The coefficient vectors $\vec{A}$ and $\vec{C}$ are calculated as follows.
(3)$$\vec{A}=2\vec{a}\cdot \vec{r_{1}}-\vec{a}$$
(4)$$\vec{C}=2\cdot {\vec{r}}_{2}$$
where $\vec{a}$ indicates the linear reduction from 2 to 0 in the iterative process, ${\vec{r}}_{1}$ and ${\vec{r}}_{2}$ are random vectors in [0, 1].

In the hunting process, α leads the group, and β and δ may sometimes participate. It is assumed that the solutions of α, β and δ obtained up to now are preserved and other search agents (including ω wolves) are obliged to update their positions according to the position of the best search agents. The equation is defined as follows.
(5)$$\vec{D_{\alpha }}=\left|{\vec{C}}_{1}\cdot \vec{X_{\alpha }}-\vec{X}\right|,\vec{D_{\beta }}=\left|{\vec{C}}_{2}\cdot \vec{X_{\beta }}-\vec{X}\right|,\vec{D_{\delta }}=\left|{\vec{C}}_{3}\cdot \vec{X_{\delta }}-\vec{X}\right|$$
(6)$${\vec{X}}_{1}=\vec{X_{\alpha }}-{\vec{A}}_{1}\cdot (\vec{D_{\alpha }}),{\vec{X}}_{2}=\vec{X_{\beta }}-{\vec{A}}_{2}\cdot (\vec{D_{\beta }}),{\vec{X}}_{3}=\vec{X_{\delta }}-{\vec{A}}_{3}\cdot (\vec{D_{\delta }})$$
(7)$$\vec{X}(t+1)=\frac{\vec{X_{1}}+\vec{X_{2}}+\vec{X_{3}}}{3}$$
where $\vec{X_{\alpha }}$, $\vec{X_{\beta }}$ and $\vec{X_{\delta }}$ indicate the positions vectors of α, β, and δ, respectively. ${\vec{D}}_{\alpha }$, ${\vec{D}}_{\beta }$ and ${\vec{D}}_{\delta }$ represent the approximate distances between the grey wolf and α, β and δ wolves. $\vec{X}(t+1)$ is the optimal position vector of prey. When the prey stops moving, the grey wolves attack the prey, ending the hunting behavior. They separate from each other in search of prey when |A| > 1 and force them to the prey when |A| < 1.

#### 2.4.2. A Discretized GWO-PLS for Variable Selection

GWO has been successfully employed for solving global optimization problems related to continuity [34,35]. However, this continuous approach is unsuitable in case of variable selection problems. Generally, the variable selection in spectroscopy can be viewed as an optimization problem of variable combination. The RMSECV generated by PLS is taken as fitness function to estimate the optimal position for wolves. The purpose of GWO variable selection is to minimize the RMSECV. The flow chart of GWO-PLS algorithm is shown in Figure 2. The detailed procedure is displayed as follows.

An initial grey wolf matrix **Q**_*n* × *j*_, where *n* and *j* are the number of wolves and spectral variables, respectively, is randomly generated first. The **q** = {q_1_, q_2_,···, q*_j_*} is a binary vector. If q*_i_* = 1, the *i*th variable is selected. If q*_i_* = 0, the variable is not selected. The PLS model is established between the variables corresponding to 1 and the target vector **y** in training set.

The position of each wolf is updated according to Equations (6) and (7). The RMSECV in discretized GWO-PLS is used to evaluate individual grey wolf positions. $\vec{X_{\alpha }}$ is the position of the fittest wolf with lowest RMSECV value. $\vec{X_{\beta }}$ is the second lowest RMSECV following $\vec{X_{\alpha }}$. $\vec{X_{\delta }}$ is the position of the wolf with the third lowest RMSECV and all other wolves are regarded as $\vec{X_{\omega }}$. The fitness value of each grey wolf is compared between the last and new iteration, storing the position of the best fitness value. After that, the positions of wolves $\vec{X}$ are discretized by the logical function.

The above steps are repeated until *t*_max_ is reached and then the optimal grey wolf vector is output. Otherwise, *t* is increased by 1 and continues to get a new fitness.

With the optimal grey wolf vector, a PLS model is established between the spectral variables corresponding to optimal grey wolf vector and target in training set. Then PLS is employed to predict the target oil contents of samples in prediction set with the same selected variables.

Matlab was used to implement the discretized GWO-PLS algorithm. In GWO, only two parameters, i.e., iteration number *t* and the wolf number *n*, need to be optimized.

#### 2.4.3. Quantitative Analysis and Model Evaluation

As a multivariate calibration technique, partial least squares (PLS) has become popularized, combining the characteristics of principal component analysis and multiple regression. The aim of PLS is to construct a linear mathematical relationship between the spectra and target contents. It is recognized as a powerful technique when dealing with strong noise, high dimensionality and a large number of collinear variables. Such a model can quantitatively analyze single oil contents in edible blend oil from a large amount of UV-Vis spectra. In this study, Monte Carlo cross validation (MCCV) [36] was applied to choose the optimal latent variables (LVs). The number of LVs was determined as 5, 5, 6 and 8 for soybean oil, sunflower oil, peanut oil and sesame oil, respectively.

To evaluate the performance of the model, the 102 samples were divided into a training set with 51 samples and a prediction set with 51 samples for model constructing and external prediction, respectively. The RMSECV, root mean squared error of prediction (RMSEP) and determination coefficients (R^2^) were employed as the evaluation criterion. Generally, better model performance should have bigger R^2^ (up to 1) and smaller RMSECV/RMSEP.

## 3. Results and Discussion

### 3.1. The Performance of Wolves

In order to study the performance variation of a wolf pack, the RMSECV of each wolf with the iteration number (*t*) was calculated in the range of 1–300. Figure 3A–D shows the RMSECV variation of each wolf for soybean oil, sunflower oil, peanut oil and sesame oil components. In the calculation, the number of wolves was fixed as 20. To display the optimization trend of wolves clearly, RMSECV values with *t* of 10, 30, 60, 100 and 300 were shown in the Figure 3A(a–e)–D(a–e). As can be seen from Figure 3A, when *t* was 10, the RMSECVs of wolves were generally high. This meant that the wolves were far away from their prey at the beginning. The RMSECVs descended greatly as *t* reached 30. When *t* was 60, the RMSECVs continued to reduce. When *t* increased to 100, it was obvious that the difference of each wolf individual became relatively small, indicating that wolves were gathering in the direction of the prey. When *t* reached 300, the RMSECV of each wolf decreased significantly compared with that of 100. Furthermore, the RMSECVs of all wolves were almost equal for the 300th iteration. A similar change rule could be obtained from Figure 3B–D for sunflower oil, peanut oil and sesame oil.

In GWO, alpha wolves have more sensitive searching ability. When they obtained the best searching results, it meant that the wolf pack had the possibility of reaching the optimal position. In Figure 3, the arrow in each sub-figure corresponded to alpha wolf. It can be seen that the positions of alpha wolves changed constantly with *t* until the best solution was found in the whole optimization process.

### 3.2. Determination of the Iteration Number

The iteration number (*t*) is a crucial parameter mentioned in the modeling process. With too a small *t*, the convergence of the algorithm would be poor, resulting in an inferior optimization ability of grey wolves. With too a large *t*, the complexity and computational time of models would be increased. Hence, *t* produced a significant influence on the prediction results. In this research, *t* ranging from 1 to 300 was investigated.

Figure 4 shows the variation of RMSECV of alpha wolves with the iteration number for soybean oil, sunflower oil, peanut oil and sesame oil in blend oil samples. From Figure 4a, it is obvious that RMSECVs were pretty large in the first place. With increase of t, the RMSECV tended to rapidly decrease until 75. After 75, the curve of RMSECV remained stable at a fixed value. The same tendency could be found in Figure 4b–d. The RMSECV maintained steady when *t* was 75, 100 and 120. Accordingly, considering the predictive ability of models, the optimal iteration number was 75, 75, 100 and 120 for soybean oil, sunflower oil, peanut oil and sesame oil, respectively.

### 3.3. Optimization of the Wolf Number

Wolves are able to defeat large prey because they communicate and assist each other. Population size is another important parameter for GWO-PLS, which has a great influence on the optimization ability and accuracy. When the value of wolf number (*n*) was too small, the calculation speed of GWO-PLS was fast. However, due to the poor diversity of the population, it was easy to fall into local optimal solution and cause premature convergence. On the contrary, the optimization efficiency would be reduced and the global search ability would be enhanced. Therefore, the value of *n* should be selected appropriately to ensure the diversity of population and search efficiency of the algorithm.

Figure 5 depicts the variation of RMSECV with the number of wolves for soybean oil, sunflower oil, peanut oil and sesame oil in blend oil samples. To investigate the relationship between the performance of wolves and population size, the *n* with an interval of 5 was changed from 5 to 100, and the RMSECVs and the running time were adopted as parameters to evaluate models. It can be seen clearly that the RMSECVs displayed a decreasing trend on the whole in Figure 5a. The RMSECVs fluctuated up and down with the increase of *n*. When the *n* was 75, the RMSECV obtained a minimum value. After *n* exceeded 75, RMSECVs fluctuated and rose. Moreover, the running time increased almost in a straight line with the increase of *n*. However, even when the *n* reached 100, the running time was less than 5 s, indicating that the GWO-PLS was very efficient. It should be noted that similar trends were also obtained for Figure 5b–d, where the RMSECV reached a minimum when the *n* was 75, 95 and 95. Therefore, 75, 75, 95 and 95 were used as the wolf numbers for soybean oil, sunflower oil, peanut oil and sesame oil in the following discussions.

### 3.4. Prediction Results

With the optimal iteration number and wolf number determined above, the GWO-PLS model could be developed and used to predict unknown samples. In order to test the predictive ability of the proposed method, 20 independent runs were performed. The number of selected variables, mean RMSEP with their standard deviation (S.D.) and mean R^2^ with their S.D. are summarized in Table 2. As comparison, the results of full-spectrum PLS, MCUVE-PLS and RT-PLS are also listed in the table. Clearly, the full-spectrum PLS model produced the worst prediction among the four methods, while MCUVE-PLS, RT-PLS and GWO-PLS all could improve the prediction of PLS. The S.D. of PLS was zero because random resampling was not involved. Although the prediction stability of GWO-PLS was not as good as, or equivalent to, other variable selection methods, it could still be used because its S.D. was small enough. GWO-PLS could produce the least average number of selected variables, the smallest mean RMSEP and biggest mean R^2^ compared with other variable selection methods. In further comparison, both the minimum and maximum number of variables for the proposed method were the smallest ones among the three variable selection methods. This indicated that GWO-PLS could produce the best prediction with least variables. The distribution of retained variables by GWO-PLS is shown in Figure 6 for the quaternary blend oil dataset. Points with vertical short bars indicated the variables retained by GWO-PLS for soybean oil, sunflower oil, peanut oil and sesame oil, respectively. The retained variables for different component oils were different, due to the different constituents in different oils, which is reasonable.

For further comparison of the predictive ability, the relationship between measured and predicted values is exhibited in Figure 7a–h for quaternary edible blend oil by PLS and GWO-PLS, respectively. It can be seen that the fitting of GWO-PLS was much better than that of PLS for the four oils. Besides, though the R^2^ for soybean oil increased slightly, the R^2^s for sunflower oil, peanut oil and sesame oil all increased to above 0.95 by GWO-PLS. It demonstrated that the GWO-PLS method, combined with UV-Vis spectroscopy, is a promising method for quantitative analysis of quaternary edible blend oil samples.

## 4. Conclusions

This study introduced discretized GWO-PLS for quantitative analysis of quaternary edible blend oils. A total of 102 edible blend oil samples were obtained by mixing soybean oil, sunflower oil, peanut oil and sesame oil together and then their UV-Vis spectra were measured. The vector composed of 1 or 0, which suggested whether the variable was selected or not, was used as the input of GWO. The RMSECV of the training set was used as the fitness function. Wolf positions were discretized by logical function. The iteration number and the number of wolves were optimized for GWO. With the optimal parameters, GWO-PLS was employed to predict the contents of single oils in quaternary blend oil samples. Results showed that GWO-PLS can provide the best predictive ability compared with full-spectrum PLS, MCUVE-PLS and RT-PLS. Thus, GWO-PLS is a feasible variable selection method for rapid quantification of quaternary edible blend oil.

## Figures and Tables

**Figure 1 molecules-27-05141-f001:**
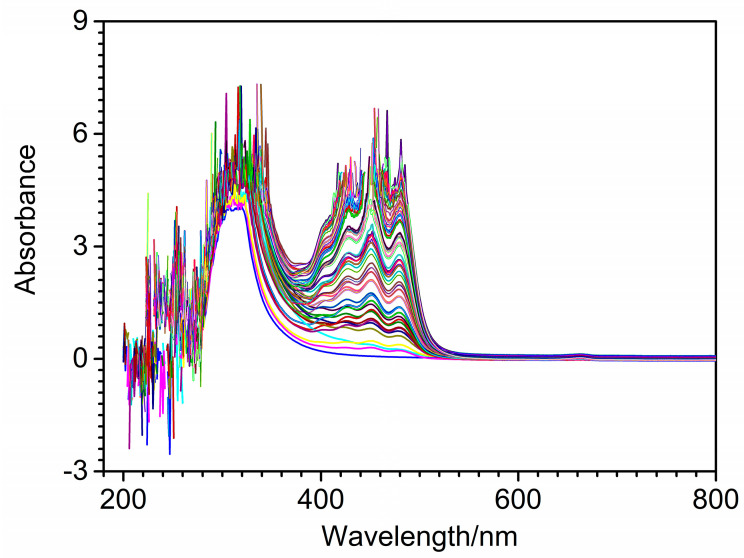
UV-Vis spectra of 102 quaternary blend oil samples.

**Figure 2 molecules-27-05141-f002:**
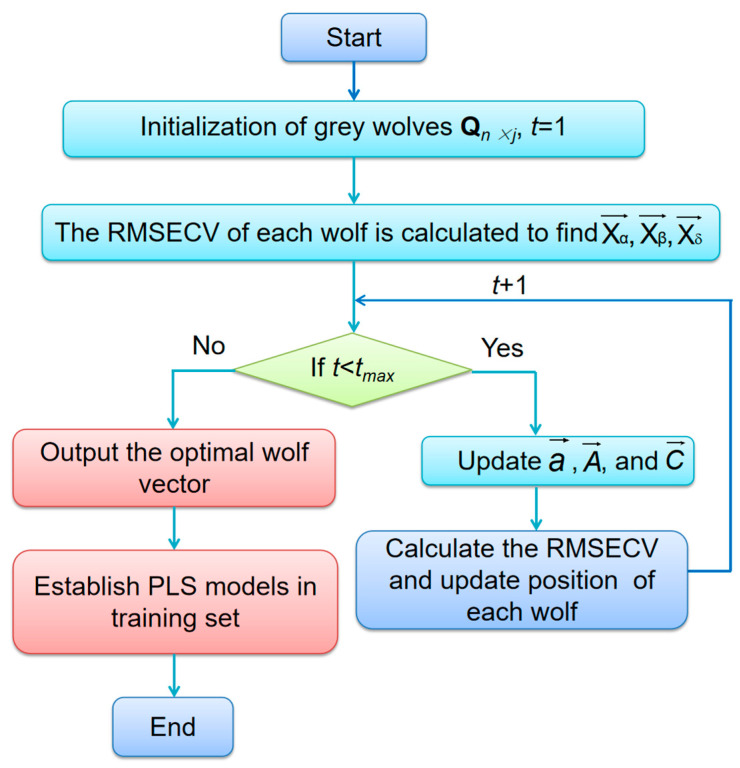
Flowchart of GWO-PLS algorithm.

**Figure 3 molecules-27-05141-f003:**
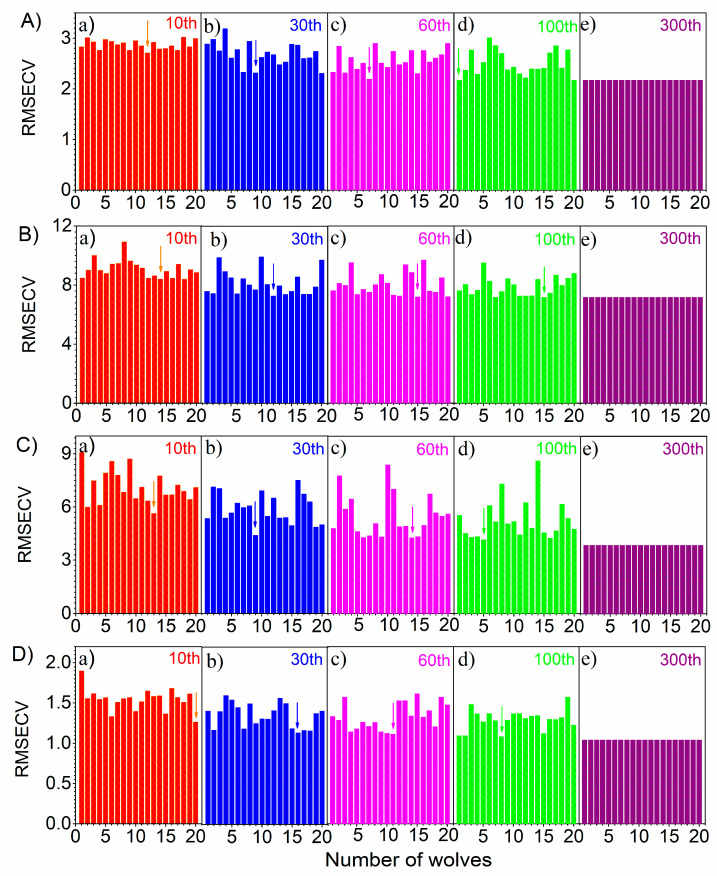
The RMSECV of 20 wolves in 10th (**a**), 30th (**b**), 60th (**c**), 100th (**d**) and 300th (**e**) iterations for soybean oil (**A**), sunflower oil (**B**), peanut oil (**C**) and sesame oil (**D**) in blend oil samples, respectively.

**Figure 4 molecules-27-05141-f004:**
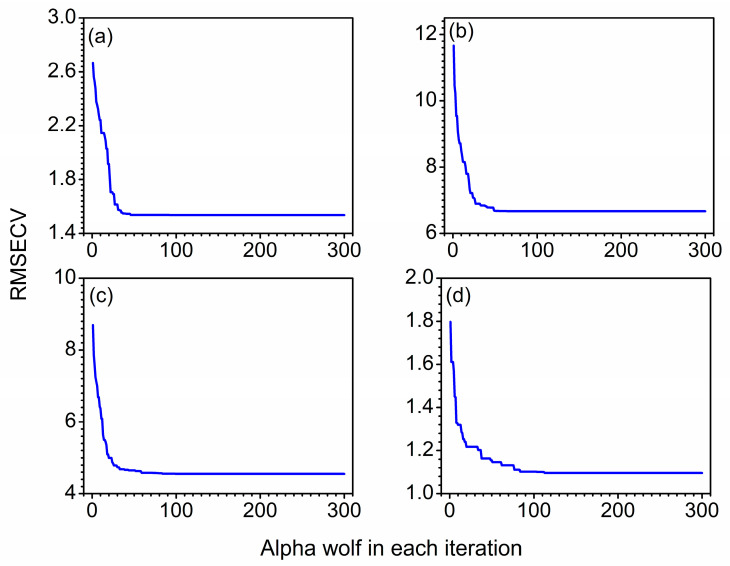
Variation of RMSECV with the number of iterations for soybean oil (**a**), sunflower oil (**b**), peanut oil (**c**) and sesame oil (**d**), respectively.

**Figure 5 molecules-27-05141-f005:**
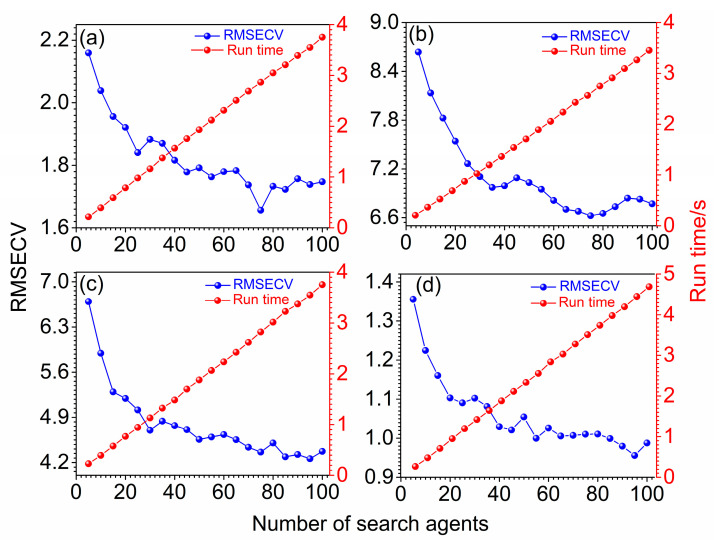
Variation of RMSECV and run time with the number of wolves for soybean oil (**a**), sunflower oil (**b**), peanut oil (**c**) and sesame oil (**d**), respectively.

**Figure 6 molecules-27-05141-f006:**
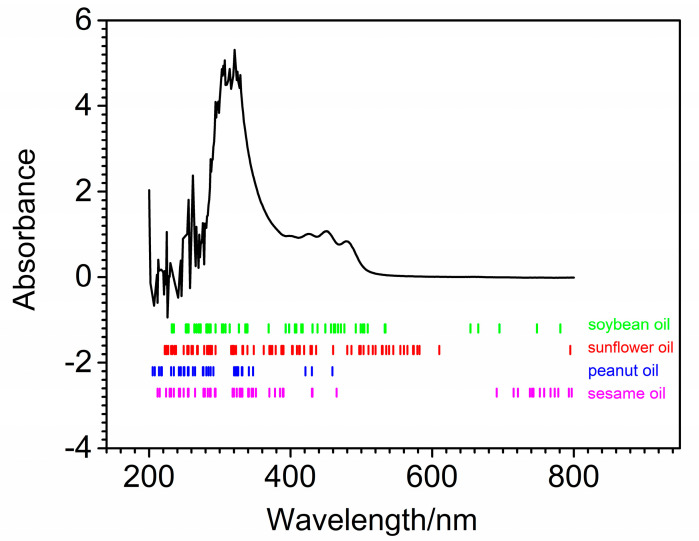
Distribution of selected variables by GWO-PLS for quaternary blend oil samples.

**Figure 7 molecules-27-05141-f007:**
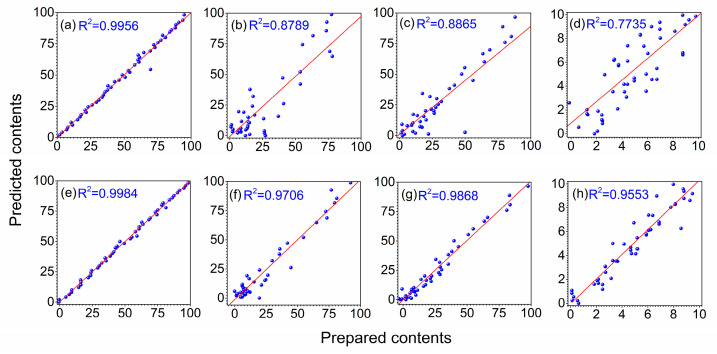
Relationship between the prepared and predicted values of the blend oil samples in prediction set by PLS for soybean oil (**a**), sunflower oil (**b**), peanut oil (**c**) and sesame oil (**d**) and by GWO-PLS for soybean oil (**e**), sunflower oil (**f**), peanut oil (**g**) and sesame oil (**h**), respectively.

**Table 1 molecules-27-05141-t001:** Concentration statistics of training and prediction sets.

Components	Training Set				Prediction Set			
	Number	Range (%)	Mean (%)	S.D. (%)	Number	Range (%)	Mean (%)	S.D. (%)
Soybean oil	51	0.0000–100.0000	50.0475	29.4469	51	0.0000–100.0000	50.0464	29.3892
Sunflower oil	51	0.0000–98.9873	21.0154	27.0065	51	0.0000–98.9790	21.0174	27.1036
Peanut oil	51	0.0000–96.6313	23.9602	25.6240	51	0.0000–96.6733	23.9459	25.5380
Sesame oil	51	0.0000–10.1021	4.9777	2.9918	51	0.0000–11.3495	4.9895	3.0449

**Table 2 molecules-27-05141-t002:** Prediction results of different methods.

Components	Methods	Number of Variables ^a^	RMSEP (S.D.) ^b^	R^2^ (S.D.) ^c^
Soybean oil	PLS	601	2.7821 (0)	0.9955 (0)
	MCUVE-PLS	383 (100–520)	2.6181 (0.0424)	0.9960 (0.0001)
	RT-PLS	235 (40–530)	2.5243 (0.0617)	0.9963 (0.0002)
	GWO-PLS	70 (50–100)	1.7845 (0.1867)	0.9981 (0.0004)
Sunflower oil	PLS	601	12.9824 (0)	0.8789 (0)
	MCUVE-PLS	125 (110–160)	10.9133 (0.0887)	0.9173 (0.0011)
	RT-PLS	111 (110–120)	10.7287 (0.2436)	0.9222 (0.0050)
	GWO-PLS	78 (43–107)	6.8456 (0.2935)	0.9678 (0.0029)
Peanut oil	PLS	601	12.3936 (0)	0.8865 (0)
	MCUVE-PLS	88 (60–140)	9.2418 (0.4848)	0.9335 (0.0072)
	RT-PLS	80 (50–150)	10.5005 (0.2259)	0.9155 (0.0038)
	GWO-PLS	64 (34–96)	4.3657 (0.2824)	0.9854 (0.0020)
Sesame oil	PLS	601	2.0009 (0)	0.7734 (0)
	MCUVE-PLS	76 (60–340)	1.8964 (0.0192)	0.7880 (0.0049)
	RT-PLS	282 (40–540)	1.9693 (0.0151)	0.7839 (0.0094)
	GWO-PLS	75 (51–124)	0.9642 (0.0542)	0.9502 (0.0061)

^a^ The average number of selected variables obtained by 20 runs, the minimal and maximal retained variables in the parenthesis, respectively. ^b^ RMSEP and S.D. is the average value and the standard deviation of 20 RMSEPs obtained by 20 runs, respectively. ^c^ R^2^ and S.D. is the average value and the standard deviation of 20 R^2^s obtained by 20 runs, respectively.

## Data Availability

Data is available from corresponding author upon request.

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
