# Peer review of "Grey Wolf Optimizer for Variable Selection in Quantification of Quaternary Edible Blend Oil by Ultraviolet-Visible Spectroscopy"

_molecules, 2022, doi:10.3390/molecules27165141_

Round 1
Reviewer 1 Report
Manuscript molecules-1829066 highlights the use of maschine learning algorhitms in order to differentiate different edible oils based on the UV-VIS spectra. The authors applied the novel technique of Grey Wolf Optimized which is nowadays applied on foodstuffs. The hypothesis driven in the present study is of great importance as the methodology proposed by the authors can be also applied on the authenticity and adulteration control of edible oils with cheaper oils. This could be a future study in collaboration.
The manuscript has been prepared according to the journal's guidelines and is in general well organized. Figures are of good quality. The authors must, however, revise the English language and provide some additional explanation on the components that absorb in the range of 350-800 nm. I have indicated the requested corrections within the attached pdf.
Based on these comments, I suggest a minor revision prior to further consideration for publication.

Author Response
Thank you very much for your affirmation of the research. Those comments are valuable and very helpful to us. We have responded the comments carefully one by one. Please see the attachment

Reviewer 2 Report
The manuscript ” Grey Wolf Optimizer for Variable Selection in Quantification of Quaternary Edible Blend Oil by Ultraviolet-Visible Spectroscopy” concerns the test of a new spectroscopic data analysis approach which could be used in the control of edible oils blends. The proposed approach has some novelty, and the problem it seeks to solve is real. The subject of the manuscript is in scope of Special Issue of Molecules "Application of Spectroscopy and Chemometrics for Authentication of Foods and Drugs", so it could be considered for publication in this issue.
However, I have some remarks that should be addressed before publication:
1. What was the repeatability of the algorithms results? And for how many independent trials it was checked? As it is a metaheuristic algorithm, this assessment of repeatability should be added.
2. What software has been used for the compilation of algorithms? Please add this to the Experimental part.
3. It would be beneficial to add the flowchart (framework) illustrating the GWO-PLS hybrid method
4. Please give more detail about the construction of training and prediction sets.
5. In my opinion, the work of Chanda et al 2017 IEEE Calcutta Conference (CALCON). doi:10.1109/calcon.2017.8280762 should be cited here, as it also concerns the application of GWO-PLS combination to the analysis of spectroscopic datas (though NIR, not UV-Vis, like in the submitted manuscript, and applied for minor component - polyphenol – in tea leaves analysis, not oils).
6. In the Experimental part it is stated that the concentration of sunflower oil was in the range of 0-98% (g/g), while in Figure 5 the relationship between prepared and predicted values is shown up to 80%. Please unify this or give an explanation why these ranges are different.
Author Response
Thank you very much for your affirmation of the research. Those comments are valuable and very helpful to us. We have responded the comments carefully one by one. Please find it in attachment.

Reviewer 3 Report
The paper presents an interesting feature selection tool for regression analysis by combining GWO and PLS for the quantification various edible oil blends. In as much as neither technique is new/novel, combining it improves the scientific knowledge and broadens the application of GWO.
1. Since GWO-PLS is being presented as a new combination technique, it is very important to know how it compares to the many available feature selection tools. Thus, a stronger case can be made for the relevance of the paper if it performs better or at least its comparable to already existing techniques. Am afraid comparing GWO-PLS and PLS may not be enough and as such will be interest to see how other FS methods implemented prior to PLS will lead to.
Highlighting the clear advantage of their FS tool also makes its usefulness much more clearer.
2. The paper needs a lot of work with regards to the writing style. There are several grammatical errors and most of the sentences can be reworded to improve clarity. Some abbreviations are also misspelt (PCR instead of PCA).
3. When references are cited, the authors should state what is to be found in that reference. (A large amount of variable selection methods are constantly emerging [19-23]).
4. On the data collection, is there a reason why the percentage w/w of sesame was only up to 10%?
5. As a feature selection tool, the features that were selected and their potential relevance to the model is also clearly missing in the manuscript.
6. In general the manuscript needs to be improved.
7. Figures have no legends.
Author Response

(The authors gave the same response as above.)

Round 2
Reviewer 2 Report
The revised manuscript is significantly improved, and can be published in Molecules with some small modifications:
1. In Table 2, in the legend, the b and c upper indexes correspond to other values (R or RMSEP) than in the column titles.
2. In the added paragraphs some English editing is needed (grammar errors).
Author Response

(The authors gave the same response as above.)

Reviewer 3 Report
Page 7 Line 11 - 13.
The statement
To evaluate the performance of the model, the samples were divided into training set with 51 samples and prediction set with 51 samples for model constructing and external prediction according to concentration sorting. Is a bit confusing.
Removing the statement "according to concentration sorting. Is a bit confusing." and adding respectively makes it a bit clearer.
To evaluate the performance of the model, the samples were divided into training set with 51 samples and prediction set with 51 samples for model constructing and external prediction, respectively.
Author Response

(The authors gave the same response as above.)
